# Phytosynthesis and Characterization of Silver Nanoparticles from *Antigonon leptopus*: Assessment of Antibacterial and Cytotoxic Properties

**DOI:** 10.3390/pharmaceutics17050672

**Published:** 2025-05-20

**Authors:** Marisol Gastelum-Cabrera, Pablo Mendez-Pfeiffer, Manuel G. Ballesteros-Monrreal, Brenda Velasco-Rodríguez, Patricia D. Martínez-Flores, Sergio Silva-Bea, Vicente Domínguez-Arca, Gerardo Prieto, Silvia Barbosa, Ana Otero, Pablo Taboada, Josué Juárez

**Affiliations:** 1Posgrado en Nanotecnología, Departamento de Física, Universidad de Sonora, Unidad Regional Centro, Hermosillo 83000, Sonora, Mexico; a213210582@unison.mx (M.G.-C.); a204200368@unison.mx (P.D.M.-F.); 2Grupo de Física de Coloides y Polímeros, Departamento de Física de Partículas, Facultad de Física, Instituto de Materiales (IMATUS), Universidad de Santiago de Compostela, 15782 Santiago de Compostela, Spain; brenda.velasco.rodriguez@usc.es (B.V.-R.); vicente.dominguez@rai.usc.es (V.D.-A.); xerardo.prieto@usc.es (G.P.); silvia.barbosa@usc.es (S.B.); pablo.taboada@usc.es (P.T.); 3Departamento de Ciencias Químico-Biológicas y Agropecuarias, Universidad de Sonora, Caborca 83600, Sonora, Mexico; pablo.mendez@unison.mx (P.M.-P.); manuel.ballesteros@unison.mx (M.G.B.-M.); 4Centro de Nanociencias y Nanotecnología, Universidad Nacional Autónoma de México, Ensenada 22860, Baja California, Mexico; 5Centro Singular de Investigación en Química Biolóxica e Materiais Moleculares (CiQUS), Universidad de Santiago de Compostela, 15782 Santiago de Compostela, Spain; 6Laboratorio de Nanostructuras Terapéuticas Magnéticas Avanzadas, Laboratorio Internacional Ibérico de Nanotecnología (INL), 4715-330 Braga, Portugal; 7Departamento de Microbiología y Parasitología, Facultad de Biología-Centro de Investigación en Salud Acuática (ARCUS), Universidad de Santiago de Compostela, 15782 Santiago de Compostela, Spain; sergio.silva.bea@usc.es (S.S.-B.); anamaria.otero@usc.es (A.O.); 8Grupo de Sistemas Biológicos e Ingeniería de Bioprocesos (Bio2Eng), Instituto de Investigaciones Marinas del Consejo Superior de Investigaciones Científicas (IIM-CSIC), 36208 Vigo, Spain

**Keywords:** phytosynthesis, silver nanoparticles, antibacterial activity, cytotoxic properties, *Antigonon leptopus*

## Abstract

**Background:** Silver nanoparticles (AgNPs) show promises as antimicrobial biomaterials with use for combating multidrug-resistant microorganisms, and they are widely used in healthcare, medicine, and food industries. However, traditional physicochemical synthesis methods often require harsh conditions and toxic reagents, generating harmful waste. The synthesis of AgNPs using plant-derived bioactive compounds offers an eco-friendly alternative to conventional methods. **Methods:** In this study, a bio-green approach was employed to synthesize AgNPs using ethanolic extracts from *Antigonon leptopus* leaves (EXT-*AL*). The synthesis was optimized under different pH conditions (5.5, 8.0, 10.0) and EXT-*AL* concentrations (10–200 μg/mL). Antibacterial activity was evaluated against *Escherichia coli* and *Staphylococcus aureus*, and cytotoxicity was assessed in HeLa, CaCo-2, T731-GFP, and HaCaT cell lines. **Results:** UV-Vis spectroscopy confirmed nanoparticle formation, with a surface plasmon resonance peak at 410 nm. Alkaline conditions (pH 10.0) favored the formation of smaller, spherical AgNPs. Characterization by DLS, TEM, and AFM revealed uniform nanoparticles with a hydrodynamic diameter of 93.48 ± 1.88 nm and a zeta potential of −37.80 ± 1.28 mV. The AgNPs remained stable in Milli-Q water but tended to aggregate in PBS, DMEM, and MHB media. Antibacterial assays demonstrated significant bactericidal activity against *Escherichia coli* and *Staphylococcus aureus* at 3.9 μg/mL (Ag⁺ equivalent). Cytotoxicity tests showed no toxicity to HeLa, T731-GFP, CaCo-2, or HaCaT cells at concentrations ≥ 7.8 μg/mL after 24 h. **Conclusions:** These findings highlight *Antigonon leptopus* extract as a sustainable and cost-effective resource for AgNPs synthesis, with strong antimicrobial properties and potential biomedical applications.

## 1. Introduction

The properties of materials at the nanoscale have sparked remarkable research activities focused on the fabrication, characterization, and applications of nanoparticles (NPs). NPs are small structures ranging from 1 to 100 nm [1,2]. At this scale, these nanosystems exhibit unique and exceptional physical, chemical, and optical properties, among others, which have garnered interest for a wide range of applications across various scientific disciplines, including biomedicine, pharmaceutics, agriculture, the food industry, reactions catalysis, electronics devices, sensors, optics, etc. [3,4,5].

Among the numerous types of nanoparticles, such as those made of silver, gold, polymeric, copper, iron, etc. [6,7], silver nanoparticles (AgNPs) have emerged as particularly promising for biomedical applications, occupying a prominent position in research [8,9]. This is due to their excellent biological activity, including antibacterial and antifungal properties, antioxidative activity, anti-tumoral and anti-inflammatory effects, and potential uses in wound dressings, drug delivery, biosensing, and biocatalysis [10,11,12,13,14].

There are many approaches to synthesizing AgNPs, including chemical reduction, photochemical and electrochemical methods, biogenic synthesis, thermal decomposition, laser ablation, gamma and microwave irradiation, etc. [14,15]. However, these conventional methodologies often rely on using chemicals, solvents, and the application of experimental conditions that are not environmentally friendly. As a result, they may produce toxic byproducts, making the resulting AgNPs potentially hazardous for their intended use in biomedical applications. In contrast, biosynthesis or green synthesis of AgNPs using natural compounds is considered an environmentally friendly alternative, as natural compounds can act as both reducing and stabilizing agents [16,17].

Additionally, green synthesis-based methods are easy to implement and safer, as plants provide a rich source of antioxidant compounds that have proven effective in the eco-friendly synthesis of bioactive AgNPs [18]. In this context, an abundance of bioactive compounds, including alkaloids, flavonoids, terpenoids, tannins, saccharides, phenols, and vitamins, along with enzymes, amino acids, and proteins, can be isolated from plants [19,20]. These biomolecules allow the synthesis of AgNPs and contribute to enhancing their stability. In this regard, numerous studies have explored the phytosynthesis (green synthesis) of AgNPs using various plant parts, such as fruits, seeds, roots, flowers, stems, leaves, and peels [21,22].

The emergence of multidrug-resistant microorganisms poses a significant global health risk [23,24], resulting in higher incidence and mortality rates, as well as increased treatment costs for infections caused by these bacteria. The rise of multidrug-resistant bacterial strains is continuously increasing due to mutations, excessive antibiotic use, and changing environmental conditions [25,26,27,28]. This is particularly critical in respiratory infections caused by antibiotic-resistant bacteria, which represent a major global health concern, especially in immunocompromised patients [29]. In this context, the ineffectiveness of many current antibiotics underscores the urgent need to explore novel alternative antimicrobials, including natural compounds, essential oils, and AgNPs.

For example, essential oils, such as carvacrol, true cinnamon, and thyme, are characterized by their bioactive, antioxidant, and antibacterial properties [30,31,32,33]. Flavonoids such as kaempferol and tannic acid show broad activity against microorganisms and have anticancer effects [33,34]. AgNPs synthesized using plant extracts have demonstrated enhanced antimicrobial activity, even against multidrug-resistant strains [35,36]. For example, Garibo et al. (2020) employed extracts of *Lysiloma acapulcensis* for the green synthesis of AgNPs and reported high-antimicrobial activity [37]. Similarly, the *Perilla frutescens* leaf extract used in synthesizing AgNPs, as Zhang et al. (2021) reported, demonstrated biological activities, such as the antibacterial, antioxidant, and anticancer effects [38]. Green synthesis using carrot extract has been shown to exhibit antibacterial activity against multidrug-resistant bacteria [39].

*Antigonon leptopus* (*AL*) is an invasive, evergreen, woody liana native to Mexico that has spread to various regions worldwide. *AL* is an important medicinal herb in Mexico [27,40]. Traditionally, it has been used in ethnomedicine to treat conditions such as pain, diabetes, skin problems, respiratory, and intestinal infections, as well as conditions like cough, bronchitis, and throat inflammation [41,42,43]. Various parts of the plant are utilized for these purposes, as they contain a complex mixture of phytochemical compounds, such as alkaloids, glycosides, flavonoids, tannins, and terpenoids, among others [43]. These compounds exhibit significant pharmacological activities, including antimicrobial, hepatoprotective, analgesic, anti-inflammatory, cytotoxic, and antidiabetic effects [41,44,45].

Notably, the antioxidant activity of phytochemicals extracted from various parts of the plant, including flowers, leaves, roots, rhizomes, or the whole plant, can be harnessed for the green synthesis of AgNPs, thereby further expanding the application of *AL*. AgNPs synthesized using natural compounds could offer advantages, such as biocompatibility and broad potential for biomedical applications. For instance, AgNPs synthesized through green approaches have been proposed as antibacterial agents. Due to their unique antibacterial properties, they have demonstrated a wide range of biological applications, including use in medical devices and cleaning agents [18,19,21].

In this study, a green synthesis method for AgNPs was optimized by varying the pH of the solution conditions and the concentration of ethanolic leaf extract of *Antigonon leptopus* (EXT-*AL*). The phytocompounds found in *AL* leaves, including alkaloids, glycosides, flavonoids, tannins, and terpenoids, are well known for their strong antioxidant properties [42,46]. For this reason, antioxidant compounds were extracted from the leaves using ethanol as an extraction solvent, which was subsequently used for the synthesis of AgNPs.

Biosynthesis of AgNPs has been widely explored as an eco-friendly alternative to conventional chemical and physical methods. However, limited research has explored how specific synthesis parameters, such as pH, extract concentration, and reaction conditions, affect the physicochemical properties of the resulting nanoparticles. These characteristics, including particle size, shape, and colloidal stability, are critical as they directly influence the biological activity and potential applications of AgNPs, particularly their antimicrobial efficacy and cytotoxicity.

To address these gaps, the present study investigates the phytosynthesis of AgNPs using EXT-*AL* and evaluates their physicochemical features, antibacterial performance, and in vitro biocompatibility. These results suggest that the EXT-*AL*-mediated AgNPs synthesis holds promise as a strategy to combat infections caused by drug-resistant bacterial strains. Given their potent antimicrobial properties, these nanoparticles could be developed into therapeutic agents for respiratory infections, especially those involving antibiotic-resistant pathogens. Potential applications include their incorporation into drug delivery platforms, inhalable formulations, or surface coating for respiratory medical devices, offering innovative solutions for the prevention and treatment of pulmonary infections.

## 2. Materials and Methods

### 2.1. Chemical Reagenents

Silver nitrate (≥99%), sodium hydroxide (NaOH), HPLC-grade ethanol, and Mueller–Hinton bacterial culture medium were purchased from Sigma-Aldrich Co. (St. Louis, MO, USA). Dulbecco’s Modified Eagle medium, L-asparagine (98%), L-arginine monohydrochloride (≥98%), L-glutamine solution (200 mM), sodium pyruvate solution (100 mM), penicillin-streptomycin solution (1000 U/1 U per mL), fetal bovine serum (FBS), trypsin-EDTA (0.25×), Alexa Fluor 647, and PBS pH 7.4 (10×) were obtained from Hyclone (Thermo Scientific, Waltham, MA, USA). The Prolong Antifade reagent with DAPI was obtained from Molecular Probes, and the CCK-8 assay kit was obtained from Gerbu Biotechnik (Heidelberg, Germany). The plant material to obtain the *Antignon leptopus* ethanolic extract (EXT-*AL*) was collected from the garden at the Universidad de Sonora Regional Center Unit (29°04′55.8″ N 110°57′37.4″ W). All other reagents were of analytical grade and/or suitable for cell culture, as appropriate, and were used as received. Milli-Q water (Millipore, Burlington, MA, USA) was used in all experiments.

### 2.2. Synthesis of AgNPs with Antignon leptopus Ethanolic Extract (EXT-AL)

The ethanolic extract of *Antigonon leptopus* leaves (EXT-*AL*) was used as a reducing agent in the synthesis of AgNPs. The *Antigonon leptopus* leaves were dried at 40 °C for 48 h. Then, 10 g of dried leaves were added to 100 mL of ethanol, and extraction was conducted for 48 h. Next, the solution was filtered (Whatman No. 1, with a pore size of 11 μm), evaporated under a low-pressure vacuum, and the ethanolic extract was stored at −20 °C.

To prepare AgNPs, the dried extract of *Antigonon leptopus* was dissolved at 5 mg/mL in ethanol (stock solution) concentration. Subsequently, different solutions of EXT-*AL* were prepared at concentrations of 10, 20, 50, 100, and 200 µg/mL in Mili-Q water previously adjusted with NaOH at pH values of 5.5, pH 8.0, and pH 10.0, which were then used to synthesize AgNPs. Afterward, silver nitrate (10 mM) was added to the EXT-*AL* solution to obtain a final concentration of 1 mM; this solution was stirred at 800 rpm for 20 min and kept in the darkroom for 48 h. Next, the AgNPs were centrifuged at 10,000× *g* for 30 min at 5 °C and washed using ultrapure water. The resulting AgNPs were stored at 4 °C, and their characteristics were determined.

### 2.3. UV-Vis Spectroscopy Analysis

UV-vis spectroscopy confirmed the formation of AgNPs by determining the presence of the localized surface plasmon resonance (LSPR) peak located at the NPs around 430 nm. The UV-Vis absorption spectra were recorded using a PerkinElmer UV-Vis spectrophotometer (Waltham, MA, USA) using a 1 cm quartz cell. For the measurements, AgNPs and EXT-*AL* were placed in a cuvette at 1:4 dilution, and the UV-Vis spectrum was recorded in the 300–800 nm range. Baseline correction was applied using Mili-Q water for EXT-*AL*, and concentrations of EXT-*AL* at each sample for baseline correction of AgNPs were employed.

### 2.4. The Ferric Ion Reducing Antioxidant Power (FRAP) Assay

The FRAP assay was performed following the protocols of Benzie and Strain; 1996 and Rao et al., 2023 [47,48]. The FRAP reagent was freshly prepared before each measurement by mixing buffer solutions of acetate (0.3 M), 2,4,6-tripyridyl-s-triazine (10 mM TPTZ in HCl (40 mM), and FeCl_3_ (20 mM in distilled water) in a 10:1:1 (*v*/*v*/*v*) ratio. The reagent mixture was incubated at 37 °C for 10 min before use. For analysis, 900 µL of FRAP reagent was mixed with 30 µL of the sample solution and 90 µL of water. The samples were incubated at 37 °C for 30 min before absorbance measurement. For spectrophotometric analysis, 200 µL of the reaction mixture was transferred to a microplate, and absorbance was recorded in the 450–650 nm range using an Agilent BioTek Epoch microplate reader.

A blank sample was prepared by omitting the sample. A calibration curve was constructed using ascorbic acid standards in the concentrations range of 0.5–5 µg/mL, and absorbance recorded at 593 nm was plotted against concentration. Prior to the assay, stock solutions of ascorbic acid and EXT-*AL* were prepared at pH 5.5 and 10.0 under conditions similar to those used for AgNPs synthesis. The reducing power of EXT-*AL* is expressed as its ascorbic acid equivalent in μg/mL.

### 2.5. Kinetics Growth Curve

The formation of AgNPs synthesized with 100 µg/mL of EXT-*AL* at pH 10.0 was monitored using a LUMIstar Omega Microplate (BMG Labtech, Aylesbury, UK; software version 3.32 R5) UV-Vis spectrophotometer at wavelengths between 300 and 600 nm. The synthesis was replicated by adding 2.7 mL of the 100 µg/mL of EXT-*AL* solution (at pH 10.0) into a 6-well microplate (Costar, Corning, CA, USA) and 300 µL of silver nitrate at 10 mM concentration. During the reduction process, the reaction mixture was stirred. After adding the silver nitrate (time 0), an initial reading was taken immediately, followed by additional readings every 20 min for 24 h. The reaction rate was calculated based on the increasing absorbance of AgNPs at 410 nm wavelength.

### 2.6. Dynamic Light Scattering (DLS)

The hydrodynamic diameter (D_h_) and the polydispersity index (PDI) of the AgNPs obtained in the presence of 100 μg/mL of EXT-*AL* were determined by DLS using a Zetasizer-Nano ZS (Malvern Instruments, Malvern, UK), equipped with a red laser of wavelength λ_o_ = 633 nm (He-Ne, 4.0 mW). Samples were diluted at 1:2 in water and placed in a disposable cell for analysis. After reaching thermal equilibrium at 25 °C, measurements in triplicate were completed. D_h_ values were obtained from DLS measurements at an incidence angle of 90°, with data analyzed using the CONTIN algorithm developed by Provencher [49]. The diffusion coefficient and the D_h_ were obtained by the Stokes–Einstein equation:(1)$$D_{h}=k_{B}\cdot \frac{T}{3\pi \eta D}$$
where D_h_ is the hydrodynamic diameter, k_B_ is the Boltzmann constant, T is the temperature, η the solution viscosity, and D is the diffusion coefficient of the particles. The software automatically determined the number of runs for each measurement.

### 2.7. Zeta Potential (ZP)

The zeta potential (ZP) of the synthesized AgNPs was measured using a Zetasizer-nano ZS (Malvern Instruments, UK). ZP was determined based on the electrophoretic mobility using the Henry equation:(2)α=ε·ζ/η
where α, ε, ζ, and *η* denote the electrophoretic mobility, permittivity of the medium, zeta potential of the particles, and viscosity, respectively. The AgNPs solutions at a 1:2 *v*/*v* ratio were filled into a folded capillary cell with two gold electrodes. Measurements were initiated after the system reached thermal equilibrium at 25 °C, and the software automatically determined the number of runs for each experimental point.

### 2.8. Transmission Electron Microscopy (TEM) and Atomic Force Microscopy (AFM)

The morphology of AgNPs was determined by TEM images captured using a JEOL JEM1011 transmission electron microscope (JEOL, Peabody, MA, USA) at an accelerating voltage of 100 kV. Samples were prepared by carefully placing a single drop of AgNPs, diluted at a 1:3 ratio, onto a copper-coated grid and allowing it to dry. ImageJ 1.53s software (National Institutes of Health, Bethesda, MD, USA) was again used for image processing. The average particle size of the synthesized AgNPs was determined by measuring the size of 300 AgNPs, and the size distribution was represented in a plotted histogram.

AFM images were acquired using an Atomic Force Microscope (JSPM-4210 model, JEOL, Akishima, Japan). The sample was prepared by placing a drop of the AgNPs, diluted at a 1:4 ratio, onto a freshly cleaved mica surface. After 60 s, the excess solution was removed with blotting paper, and the sample was left to dry for 1 h at room temperature. Images were obtained in non-contact mode using an HQ: NSC15/AI BS cantilever. The images were analyzed using WSxM 5.0 software (Julio Gómez Herrero & José María Gómez Rodríguez) and ImageJ 1.53s software (National Institutes of Health, Bethesda, MD, USA).

### 2.9. Fourier-Transform Infrared Spectroscopy (FTIR)

FTIR spectra were recorded at the end of the synthesis process to confirm the formation of a complex AgNPs@EXT-*AL*, where the presence of the EXT-*AL* compounds adsorbed onto the surface of the AgNPs core, acting as stabilizing agents, forming an organic shell around the AgNPs. FTIR spectra were performed using a Perkin-Elmer spectrometer (Waltham, MA, USA). The EXT-*AL* and AgNPs were previously lyophilized, and approximately 30 mg of each sample was placed in a diamond ATR Crystal. The measurements were carried out using the ATR method, with a resolution of 4 cm^−1^ in the range from 4000 to 400 cm^−1^.

### 2.10. Thermogravimetry Analysis (TGA)

The thermal behavior of EXT-*AL* and AgNPs was evaluated by thermogravimetric analysis (TGA) using a TGA 55 thermogravimeter analyzer (TA Instruments, New Castle, DE, USA). Samples ranging from 3 to 10 mg were heated in a platinum pan from 40 to 1000 °C at a 10 °C/min rate under an N_2_ atmosphere.

### 2.11. X-Ray Diffractometry (XRD)

X-ray diffraction (XRD) was used to determine the crystallinity of the AgNPs. The XRD pattern of AgNPs was obtained using a Bruker D8 Advance diffractometer (40 kV, 40 mA, theta/theta) equipped with a sealed Cu X-ray tube CuKα1 (λ = 1.5406 Å) and a LYNXEYE XE-T detector. Diffractograms were obtained in the angular range of 3 < 2θ < 60, with steps of 0.02 and a 2-s dwell time per step. The XRD-diffractograms were compared with the JCPDS-ICDD PDF database. The average crystallite size was determined using Scherrer’s equation:(3)D=0.94λ/βcosθ
where D is the crystallite diameter, λ is the wavelength of X-ray, θ is the Bragg’s angle in radians, and β is the FWHM of the peak in radians.

### 2.12. Stability Assay

The stability of AgNPs was assessed by measuring the D_h_ and zeta potential at different time intervals. The AgNPs were washed and recovered by centrifugation at 10,000× *g* for 30 min at 5 °C. The resulting AgNPs pellet was resuspended in various aqueous media, including Mili-Q water, PBS at pH 5.5 and 7.2, Mueller–Hinton broth (MHB), and Dulbecco’s modified eagle media (DMEM). Measurements were taken at 0, 1, 2, 3, 4, 6, 7, 8, 9, and 10 days.

### 2.13. Inductively Coupled Plasma-Mass Spectrometry (ICP-MS)

The Ag^+^ concentration in solution was determined using inductively coupling plasma mass spectrometry (ICP-MS) with a Varian 820-MS instrument (Agilent Technologies, Santa Clara, CA, USA). Briefly, 1 mL of AgNPs was dissolved in 0.3 mL of HCl (37% (*v*/*v*)) and 0.1 mL of HNO_3_ (70% (*v*/*v*)). The resulting solutions were then diluted with deionized water to a final volume of 2 mL. The intensity of the emission wavelength was measured and compared to Ag^+^ standard solution. The percentage yield of Ag^+^ was calculated using the following equation:(4)$$\%\;AgNPs\;yield=\left[\frac{{Ag}^{+}\;quantified\;of\;sample\;by\;ICP-MS}{{Ag}^{+}quantified\;from\;AgNO_{3}\;by\;ICP-MS}\right]\cdot 100$$

### 2.14. Antibacterial Activity of AgNPs

The antibacterial activity of EXT-*AL* and AgNPs was evaluated using the microdilution plate method following Clinical Laboratory Standard Institute (CLSI) protocols (document M07-A10) [50], with *Escherichia coli* (ATCC 25922) and *Staphylococcus aureus* (ATCC 25923) as bacterial models. For the EXT-*AL*, 100 µL of each concentration (37.5, 75, 150, 300, and 600 µg/mL) was added to a 96-well plate (Costar, Corning), followed by the addition of 10 µL of bacterial inoculum (5 × 10^5^ CFU/mL) prepared in Mueller–Hinton broth (MHB) to each well.

Similarly, AgNPs were resuspended in MHB and diluted to concentrations of 0.975, 1.95, 3.9, 7.8, 15.6, and 31.2 µg/mL (serial dilution based on the primary value determined by ICP-MS). Subsequently, 100 µL of these AgNPs’ suspensions were dispensed into 96-well plates (Costar, Corning), followed by the addition of bacterial inoculum under similar conditions. The plates were incubated at 37 °C for 24 and 48 h, and bacterial viability was determined by measuring the absorbance at 620 nm using a microplate reader (Multiskan GO, Thermo Scientific, Waltham, MA, USA). The absorbance of EXT-*AL* resuspended in MHB without bacterial inoculum was used as a reference. Antibacterial activity was expressed as a percentage of viability.

### 2.15. Cytotoxicity Assays of AgNPs

The cytotoxicity of AgNPs was evaluated in vitro using the CCK-8 cytotoxicity assay, using human cervical adenocarcinoma cells (HeLa) and human colon adenocarcinoma (CaCo-2) provided by Cell Biolabs (San Diego, CA, USA); mouse astrocytes (T731-GFP) kindly donated by Prof. J.A. Costoya (University of Santiago, Spain); and human epidermal keratinocyte cells (HaCaT) purchased from Sigma Aldrich (St. Louis, MO, USA). Cells were cultured under standard conditions (37 °C, 5% CO_2_) in Dulbecco’s Modified Eagle Medium (DMEM), supplemented with 10% (*v*/*v*) FBS and 1% (*v*/*v*) of penicillin/streptomycin, sodium pyruvate, and nonessential amino acids.

A 100 μL suspension of viable cells (5 × 10^3^ cells/mL) from each cell line was added to each well of a flat bottom 96-well microtiter plate and incubated for 24 h under standard culture conditions. Afterward, DMEM was replaced with 100 μL of the different AgNPs solutions (0.975, 1.95, 3.9, 7.8, 15.6, and 31.2 µg/mL) diluted in DMEM, and cells were incubated at 37 °C for 24 and 48 h. At the same time, wells containing different concentrations of AgNPs without cells served as negative controls. At the end of the incubation period, cells were washed three times with PBS (pH 7.4), and 100 μL of fresh culture medium containing 10% (*v*/*v*) of CCK-8 reagent was added, followed by a 2 h incubation at 37 °C. Cell metabolic activity was measured by analyzing the optical density of the soluble formazan crystals at 450 nm using a UV-vis microplate absorbance reader (Bio-Rad, Hercules, CA, USA). The percentage of cell viability was calculated using the following equation:(5)$$\%\;Cell\;viability=\left[\frac{mean\;absorbance\;of\;test}{mean\;absorbance\;at\;control\;\left(untrated\;cells\right)}\right]\cdot 100$$

### 2.16. Statistical Analysis

All experiments were performed in triplicate. Data were analyzed using two-way ANOVA, followed by Tukey’s multiple comparisons test using GraphPad Prism 9.0 software for Windows. Experimental values were expressed as mean ± standard deviation (SD), with *p* < 0.05 considered statistically significant.

## 3. Results

### 3.1. Synthesis and Characterization of AgNPs

#### 3.1.1. Optimization of AgNPs Synthesis

The pH values of 5.5, 8.0, and 10.0 were selected based on the known pKa range of phenolic compounds present in EXT-*AL*, which typically lie between 8.0 and 10.0. At pH 5.5, most phenolic hydroxyl groups remain protonated, limiting their electron-donating capacity and, consequently, their ability to reduce Ag^+^ ions. In contrast, at pH 8.0 and especially at pH 10.0, increased deprotonation enhances the reducing power of compounds and improves nanoparticle stabilization. Therefore, these pH conditions were systematically evaluated to explore how the ionization state of phenolics influences the efficiency of AgNPs and the stability of the resulting colloidal suspension.

Figure 1 shows the UV-Vis absorption spectra recorded for AgNPs synthesized under various pH conditions (5.5, 8.0, and 10.0) in the presence of different concentrations of EXT-*AL* (10, 20, 50, 100, and 200 μg/mL). The EXT-*AL* (green line) shows a broad and irregular adsorption band ranging from 345 to 520 nm, corresponding to the presence of polyphenolic compounds in the EXT-*AL* [41,44].

The antioxidant activity of EXT-*AL* compounds was evaluated by FRAP test at pH 5.5 and 10.0. At an acidic pH (5.5), the reduction potential of EXT-*AL* compounds was relatively low (0.538 ± 0.004 μg/mL equivalent to ascorbic acid), whereas at pH 10.0, the antioxidant activity was significantly higher, reaching approximately 1.160 ± 0.005 μg/mL in ascorbic acid equivalents.

The UV-Vis peak around 430 nm, corresponding to the localized surface plasmon resonance (LSPR) band for AgNPs, confirms the successful formation of the particles (Figure 1) under these experimental conditions. The shape and intensity of the LSPR band provide valuable insight into the size, shape, and concentration of AgNPs [12]. For example, a broad and asymmetrical band suggests relatively large nanoparticle sizes and polydisperse and/or anisotropic nanoparticles. In contrast, a narrow band indicates a monodispersed pool of spherical nanoparticles [51].

In this context, the UV-Vis absorption spectra recorded for AgNPs synthesized at pH 5.5 (Figure 1A) show a characteristic band indicative of spherical nanoparticles; however, the broadness of the band suggests that the AgNPs are polydisperse. Additionally, the progressive increase in absorption intensity at higher EXT-*AL* concentrations indicates an increased formation of AgNPs.

On the other hand, the UV-Vis spectra recorded for AgNPs synthesized at pH 8.0 (Figure 1B) showed broad, low-intensity absorption bands at EXT-*AL* concentrations of 10 and 20 µg/mL, followed by a sudden increase in intensity at the EXT-*AL* concentrations of 50, 100, and 200 µg/mL. However, the broad width and asymmetrical shape of the bands persisted, indicating polydisperse and polymorphic AgNP populations. For AgNPs synthesized at pH 10.0 (Figure 1C), a low-intensity LSPR band was observed at EXT-*AL* concentrations of 10, 20, and 50 µg/mL. These absorption bands were well-defined, and their width significantly decreased at an EXT-*AL* concentration of 50 µg/mL, suggesting the formation of spherical nanoparticles with low polydispersity and low polymorphism. Moreover, the LSPR band observed for AgNPs synthesized at pH 10.0 with 100 µg/mL of EXT-*AL* was stronger and more sharply defined, indicating the formation of spherical AgNPs in high yield.

Based on these findings, AgNPs synthesized with 100 µg/mL of EXT-*AL* at pH 10.0 were chosen for further characterization, including size and zeta potential analysis.

#### 3.1.2. Particle Size and Zeta Potential Measurements

Table 1 shows the hydrodynamic diameter (D_h_), polydispersity index (PDI), and zeta potential (ZP) values determined for synthesized AgNPs at pH 10.0. The average size and ZP of AgNPs varied based on the amount of EXT-*AL* (from 10 to 50 µg/mL) used in the synthesis. The AgNPs’ size progressively increased from 95.10 to 130.90 nm, while their ZP values remained negative, without significant variations. Interestingly, the size of AgNPs synthesized with 100 µg/mL EXT-*AL* decreased to 93.48 ± 1.88 nm, with a ZP value of −37.80 ± 1.28 mV. In contrast, the AgNPs synthesized at the highest ETX-*AL* concentration (200 µg/mL) had a size of 267.8 ± 7.92 nm and a ZP value of −26.60 ± 0.90 mV.

One-way ANOVA revealed statistically significant differences in the hydrodynamic diameter (Dh), polydispersity index (PDI), and zeta potential (ZP) of AgNPs synthesized with different concentrations of EXT-*AL* (*p* < 0.0001). Tukey’s post hoc test confirmed statistically significant differences between groups (*p* < 0.05), except for the comparison between 10 and 20 µg/mL for ZP (*p* = 0.5829).

The smallest Dh (93.48 ± 1.88 nm), lowest PDI (0.25 ± 0.01), and most negative ZP (−37.42 ± 1.28 mV) were observed at 100 µg/mL, indicating greater uniformity and colloidal stability. In contrast, 200 µg/mL produced larger AgNPs (267.8 ± 7.92 nm) with lower stability (−26.12 ± 0.90 mV). These results suggest that 100 µg/mL is the optimal concentration for producing small, monodisperse, and electrostatically stable AgNPs.

These results are consistent with the findings from UV-Vis spectroscopy; AgNPs synthesized at 100 µg/mL exhibited an intense and narrow LSPR absorption band (Figure 1C, red line), indicating the formation of spherical and monodispersed AgNPs. Conversely, the low absorption intensity of the LSPR observed at EXT-*AL* concentrations other than 100 µg/mL (Table 1) suggests poor stability and low yield, accompanied by precipitation of AgNPs upon incubation. Therefore, AgNPs synthesized using 100 µg/mL of EXT-*AL* were selected for further analysis, as they demonstrated optimal stability and yield.

### 3.2. Transmission Electron Microscopy (TEM) and Atomic Force Microscopy (AFM) Analyses

TEM and AFM were used to characterize the size and morphology of AgNPs synthesized using100 µg/mL EXT-*AL* at pH 10.0 (Figure 2). TEM images (Figure 2A) confirmed the predominantly spherical morphology of the AgNPs. Size distribution analysis (Figure 2B) revealed a bimodal distribution size, with one population centered around 8 nm and another around 15 nm, indicating size heterogeneity within the range from 4 to 20 nm.

AFM images (Figure 2C) also supported the spherical morphology of synthesized AgNPs. However, the particle sizes observed by AFM were significantly larger (Figure 2D), with an average range of 80–120 nm, closely matching the results obtained from DLS measurements (Table 1).

These discrepancies in particle size arise from the fundamental differences between the techniques. AFM measures surface topography, capturing the surrounding organic layer derived from EXT-*AL* adsorbed onto the AgNPs surface. In contrast, TEM provides high resolution images of the metallic core, as the organic coating is typically transparent to the electron beam.

### 3.3. Kinetic Growth of AgNPs

The growth kinetics of AgNPs resulting from the reduction of Ag^+^ ions in the presence of EXT-*AL* (100 µg/mL) at pH 10.0 was monitored by UV-Vis spectroscopy. Figure 3A shows the evolution of the absorption band in the range 325–600 nm throughout the reaction. As the reaction progresses, the width of the adsorption band decreases while its intensity increases, with a blue shift of its λ_max_ from 430 nm to 410 nm by the end of the reaction. The absorption intensity at the λ_max_ was plotted versus reaction time, and the resulting plot displayed exponential growth behavior (inset Figure 3A). The apparent rate constant (*k*_obs_) for the formation of AgNPs was determined from the slope of the plot of $\ln\left(A_{\infty }-A_{t}/A_{\infty }-A_{0}\right)$ vs. time (Figure 3B), and the *k*_obs_ value is 2.52 × 10^−3^ min^−1^.

### 3.4. X-Ray Diffraction (XRD)

XRD analysis was performed to assess the crystallinity of Ag NPs synthesized using 100 μg/mL EXT-*AL* at pH 10.0 (Figure 4). The diffraction pattern shown in Figure 4A confirms the polycrystalline nature of the AgNPs formed under these conditions. Prominent peaks at 2θ 38.1° and 44.1° (Figure 4B) correspond to the (111) and (200) crystal planes of metallic silver, respectively, features that are characteristic of a FCC structure (ICCD PDF file No: 98-005-3761) [52]. The crystallite size calculated for the (111) plane was approximately 12.3 nm [53], which is consistent with the metallic core size observed in TEM images.

In addition to the main silver peaks, minor diffraction signals were detected that may correspond to inorganic by-products formed during the synthesis process, such as chlorargyrite, which exhibit a cubic crystal structure (ICCD file No: 98-005-6538) [54].

### 3.5. Fourier Transform Infrared (FT-IR) Spectroscopy

FT-IR spectroscopy was used to decipher the functional groups of the molecules responsible for the coating and stabilization of AgNPs. The FT-IR spectrum confirms the presence of organic compounds from EXT-*AL* in the AgNPs’ samples (Figure 5). These organic components contribute to the stability of the AgNPs [55].

In Figure 5, the main spectral signals of EXT-*AL* are observed at 3317, 2923, 1738, 1606, 1357, and 1058 cm^−1^, while the corresponding FTIR signals for AgNPs show significant peaks at 3315, 2918.5, 1722, 1603.5, 1375, and 1058 cm^−1^. The broad, intense bands located around 3315 and 3317 cm^−1^ are associated with the stretching of -OH groups present in polyphenol compounds, while the peaks at 2923 and 2918 cm^−1^ correspond to C-H and CH_2_ vibrations of aliphatic hydrocarbons, respectively [55,56,57]. The signals in 1722 and 1738 cm^−1^ are associated with the C=O stretching of carbonyl groups found in phenolic compounds. Peaks at 1603 and 1606 cm^−1^ are assigned to asymmetric C=O and aromatic C=C stretching vibrations. Additionally, signals around 1357, 1375, and 1058 cm^−1^ are attributed to heterocyclic compounds containing C-O-C groups [58,59,60].

### 3.6. Thermogravimetric Analysis (TGA)

Thermogravimetric analysis (TGA) was used to evaluate the mass loss of organic compounds as a function of temperature changes during the test. Figure 6A shows the TGA plot (brown line) of EXT-*AL*, indicating the mass loss due to water evaporation and the decomposition of polyphenolic compounds.

The three principal peaks observed in the Derivative Thermogravimetry (DTG, line green) curve at 191, 280, and 351 °C correspond to the melting and decomposition of polyphenolic compounds. Subsequently, the organic compounds undergo calcination, resulting in the maximum mass loss in the sample at a higher temperature of 400 °C.

Figure 6B shows the TGA curve (blue line) of AgNPs, along with the corresponding DTG (red line), highlighting two distinct stages of thermal degradation. The first degradation stage occurs between 150 °C and 450 °C, resulting in a mass loss of approximately 20%. This loss is attributed to the decomposition of the organic coating derived from EXT-*AL*, which consists of phenolic acids, flavonoids, and carbohydrates [61,62]. Comparable thermal behavior has been reported for glucose-coated AgNPs, which exhibited a 24% mass loss at 450 °C [36]. Similarly, AgNPs synthesized using green tea phytochemicals demonstrated a 27% mass loss within the same temperature range [36,63].

A second degradation was observed between 550 and 900 °C, accounting for an additional 35% mass loss. This stage is attributed to the calcination of residual organic compounds present in AgNPs. Then, the remaining mass corresponds to silver content, consistent with previous findings reported in the literature [64].

### 3.7. Stability of AgNPs

The stability of AgNPs synthesized with 100 μg/mL EXT-*AL* at pH 10.0 was evaluated in different media, including Mili-Q water, PBS buffer (at pH 7.2 and 5.5), DMEM, and MHB. Figure 7A,B show the size and ZP of AgNPs in the tested media.

In Mili-Q water, the size of AgNPs progressively decreases from 93 to 85 nm over 10 days. In this regard, the surface charge of AgNPs changes from −38 mV to −32 mV during this period, indicating a strong electrostatic repulsion that supports the aqueous stability of the nanoparticles [65].

In PBS buffer at pH 7.2, the size of AgNPs increased from 62 to 118 nm, while the ZP value decreased from −29 mV to −45 mV. In contrast, at pH 5.5, the AgNPs size increased from 56 nm to 84 nm at pH 5.5, with the ZP value decreasing from −31 mV to −36 mV.

In DMEM culture medium supplemented with fetal albumin, the size of AgNPs increased due to the protein adsorption onto their surface, forming large aggregates [66]. The size of AgNPs recorded by DLS was around 170 nm with a ZP of −20 mV, remaining almost constant over the 10-day incubation period. This behavior differs from that observed in the MHB culture medium, where the size increased from 150 nm to 300 nm after 10 days, while the ZP values rose from −32 mV to approximately −14 mV.

### 3.8. Ag^+^ Content in AgNPs by ICP-MS

The equivalent amount of Ag^+^ contained in AgNPs, synthesized with 100 µg/mL of EXT-*AL* at pH 10.0, was determined by ICP-MS. According to the ICP-MS data, the concentration of Ag^+^ was 32.1 ± 0.6 μg/mL. This value corresponds to an AgNPs yield of 36.0 ± 0.7% based on the initial amount of AgNO_3_ used in the synthesis process.

### 3.9. Antibacterial Activity of EXT-AL and AgNPs

Figure 8 shows the effect of EXT-*AL* and AgNPs on the viability of *S. aureus* and *E. coli* strains. EXT-*AL* alone exhibited poor antimicrobial activity (Figure 8A), even at the highest concentration tested (600 μg/mL), where the survival rates for both *S. aureus* and *E. coli* remained above 70%.

In contrast, AgNPs significantly reduced the viability of both bacterial strains (Figure 8B), with only 3.9 μg/mL of nanoparticles (equivalent to Ag^+^) required to completely inhibit their growth. As observed, the Gram-positive strain (*S. aureus*) was more susceptible to the antibacterial action of AgNPs than Gram-negative bacteria (*E. coli*).

### 3.10. Inhibition of Bacterial Growth by AgNPs

To evaluate the effect of AgNPs on the viability of *S. aureus* and *E. coli*, the kinetic growth of both bacteria was analyzed in the presence of different concentrations of AgNPs. Figure 9 illustrates the relationship between optical density and bacterial viability over incubation time. In general, it can be observed that the lag phase of bacterial growth was extended by the presence of AgNPs, 0.975 and 1.95 µg/mL of Ag^+^ equivalent with a more prolonged lag phase at the higher concentration of 1.95 µg/mL of Ag^+^ equivalent, thus suggesting a clear inhibitory effect [19]. After this phase, bacterial growth resumed. On the other hand, bacterial growth was completely inhibited when *S. aureus* and *E. coli* were incubated with ≥3.9 µg/mL of AgNPs.

For *S. aureus*, the lag phase under normal conditions, that is, in the absence of particles, was 5 h (untreated cells in Figure 9A), while in the presence of AgNPs equivalent to 0.975 and 1.95 µg/mL of Ag^+^, the lag phase increased to 6 and 10 h, respectively. In contrast, *E. coli* showed a lag phase of 3 h under normal conditions (Untreated cell in Figure 9B), which was extended to 4 h and 8 h in the presence of AgNPs at 0.975 and 1.95 µg/mL of Ag^+^, respectively. Nevertheless, both bacteria survived, reaching optical density values comparable to those of untreated controls for 0.975 µg/mL Ag^+^ equivalent after 24 h and of 1.95 µg/mL of Ag^+^ equivalent after 48 h, respectively. Interestingly, AgNPs exhibited high toxicity against both *S. aureus* and *E. coli* at an Ag^+^ equivalent concentration of 3.9 µg/mL, with no evidence of bacterial growth observed after 48 h of incubation.

### 3.11. Cytotoxicity Effect of AgNPs on HeLa, T731-GFP, CaCo-2, and HaCaT Cells

Figure 10 shows the cell viability after 24 and 48 h of incubation in the presence of different concentrations of AgNPs. The potential biocompatibility of AgNPs was evaluated in vitro by conducting cellular viability assays on model cell lines, including HeLa cells (cervical cancer), T731-GFP cells (mouse astrocytes), CaCo-2 (human colon carcinoma epithelial cells), and HaCaT (human epidermal keratinocyte). These cell models were selected to represent different metabolic profiles. HeLa cells have a high metabolic rate [67,68], while T731-GFP and Caco-2 cells have a very low metabolic rate [69,70,71,72,73], and HaCaT cells show a normal metabolism [73].

According to Figure 10, HeLa cells in the presence of AgNPs within the concentration range of 0.975 to 15.6 µg/mL Ag^+^ equivalent maintained approximately 100% of viability after 24 h of incubation. However, AgNPs were cytotoxic to HeLa cells at 31.2 µg/mL, reducing viability to 30% within the same incubation period.

For T731-GFP cells (Figure 10), the viability after 24 h of incubation was constant at approximately 100% for AgNPs concentrations below 7.8 µg/mL. However, at higher concentrations, cell viability decreased, to 55% at 15.6 µg/mL and 5% at 31.2 µg/mL. After 48 h of incubation, the AgNPs became toxic to T731-GFP cells at concentrations of 3.9 µg/mL and above.

CaCo-2 cells (Figure 10) were more susceptible to AgNPs during 24 h and 48 h of incubation. After 24 h, CaCo-2 viability decreased to 65% and remained constant across the AgNPs concentration range from 0.975 to 3.9 μg/mL. At higher AgNPs concentrations of 7.8, 15.6, and 31.2 μg/mL, cell viability decreased to 65%, 45%, and 15%, respectively. After 48 h of incubation, CaCo-2 viability decreased to 75% at 1.95 μg/mL, while 47% viability was observed across all higher AgNPs concentrations.

Finally, for HaCaT cells (Figure 10) viability remains constant across the AgNPs concentration range of 0.975 to 7.8 μg/mL. However, at concentrations beyond this range, viability decreased to 75% at 15.6 μg/mL of Ag^+^ equivalent and was lethal at 31.2 μg/mL. After 48 h of incubation, HaCaT cell viability showed a similar trend, with 100% viability at 0.975 to 3.9 μg/mL concentrations. Viability decreased to 75%, 60%, and 2% at 7.8, 15.6, and 31.2 μg/mL, respectively.

## 4. Discussion

In this study, *EXT-AL* was used as both a reducing and stabilizing agent for the synthesis of AgNPs. The observed changes in the broadness and intensity of the UV-Vis absorption spectrum of *EXT-AL* across different pH levels suggest variation in the protonation states of the polyphenolic compounds present in the extract [74,75,76]. These protonation changes are key in the successful reduction of Ag^+^ to Ag^0^ and the subsequent formation of AgNPs.

The synthesis of AgNPs was favored under alkaline conditions (pH 10.0), where electron donation from EXT-*AL* compounds to Ag^+^ was enhanced, promoting the formation of monodisperse AgNPs. This was evidenced by the stretched and well-defined absorption band [19,59,74,75,76,77]. The increased reducing capacity of EXT-*AL* at pH 10.0, confirmed by the FRAP assay, further supports this finding, indicating a stronger reduction potential compared to *EXT-AL* at pH 5.5 and 8.0. Physical characteristics, such as particle size, morphology, surface charge, and colloidal stability, are critical for the biological performance of nanoparticles. Therefore, the formulation synthesized at pH 10.0 with 100 μg/mL EXT-*AL* was chosen as optimal, yielding stable and spherical AgNPs. Similar findings were reported by Leyu et al. (2023), who synthesized AgNPs using *Parthenium hysterophorus* extract at an optimal pH condition of 10.0 [51].

The interaction of AgNPs with biological systems is highly dependent on their colloidal stability and dispersion in physiological environments. To predict their behavior and performance in subsequent biological assays, the stability of AgNPs was evaluated in water, PBS, DMEM, and MHB.

In PBS (pH 7.2 and 5.5), the observed increase in particle size and changes in ZP suggest that phosphate ions induce nanoparticle aggregation by screening surface charges, thereby reducing repulsive forces [78]. In DMEM, the larger hydrodynamic size is primarily attributed to the adsorption of serum proteins, such as fetal bovine serum albumin, onto the AgNPs surface, leading to the formation of large aggregates [66]. Similarly, in MHB, the high ionic strength resulting from Na⁺, K⁺, and Ca^2^⁺ destabilizes the colloidal system and promotes significant aggregation over time [35].

Although aggregation may enhance sedimentation and reduce the availability of dispersed AgNPs, thereby limiting their effective interactions with biological targets, the synthesized nanoparticles still showed acceptable stability in Milli-Q water, PBS, and DMEM. This is particularly relevant for biomedical applications.

The antibacterial activity of EXT-*AL* and AgNPs was evaluated against model strains of *E. coli* and *S. aureus*. The antibacterial activity of EXT-*AL* suggests that the concentration of EXT-*AL* presents potential for antibacterial activity at a low concentration level. However, the AgNPs showed strong antibacterial effects in a concentration-dependent manner, with a minimum inhibitory concentration (MIC) at 3.9 µg/mL against both strains. These are consistent with previous reports on plant-mediated AgNPs synthesis [57,58,60,61,62,63]. For example, Barabadi et al. synthesized AgNPs using *Zataria multiflora* and *Pimpinella anisum* extracts, reporting MIC values of approximately 4 µg/mL against *S. aureus* [79,80]. The antibacterial mechanism of AgNPs is multifaced; it could disrupt the bacterial cell wall, interact with essential biomacromolecules such as proteins, DNA, and RNA, thereby affecting the cellular metabolism, or bind to the bacterial cell wall, where it could disrupt the cell wall, induce oxidative stress, and eventually, cause bacterial cell death [37,81].

In terms of biocompatibility, AgNPs demonstrate selective cytotoxicity depending on the human cell line and exposure duration. For instance, after 24 and 48 h of incubation with 31.2 µg/mL AgNPs, cell viability decreased to 31–5% in HeLa cells and 21–33% in CaCo-2 cells. These findings are consistent with Jadhav et al. (2018), who observed a 65% reduction in HeLa cell viability after 24 h of exposure to 39.31 µg/mL AgNPs [82]. In a similar study, Kumar et al. (2024) reported cytotoxic effects on HeLa cells at 40 µg/mL after 48 h [83].

AgNPs exhibited marked cytotoxic toward T731-GFP and HaCaT cells, with both cell lines showing minimal viability after 24 h of exposure to 31.2 µg/mL. Notably, T731-GFP cells demonstrated partial recovery after 48 h, reaching approximately 24% viability. In contrast, HaCaT cells remained sensitive with limited recovery over the same period. Rolim et al. (2019) reported significant cytotoxicity effect only at concentrations exceeding 50 µg/mL [36], suggesting that the AgNPs synthesized in this study exert stronger cytotoxicity at comparatively lower concentrations.

These results highlight the promising potential of EXT-*AL*-derived AgNPs as antimicrobial biomaterials, particularly for localized biomedical applications where precise dose control is feasible. Nevertheless, the observed time-dependent cytotoxicity underscores the need to carefully optimize both concentration and exposure duration. Comprehensive biocompatibility assessments using advanced in vitro and in vivo models are essential to establish a robust safety profile.

In addition, further studies are necessary to evaluate the therapeutic efficacy of these AgNPs in complex models of bacterial infection, particularly those involving clinically relevant pathogens with well-characterized virulence and antibiotic resistance profiles. Such investigations will be critical in determining the clinical feasibility of EXT-*AL*-based AgNPs for antimicrobial applications, particularly in the treatment of respiratory infections.

## 5. Conclusions

In summary, this study presents a simple, reproducible, and cost-effective method for synthesizing antibacterial AgNPs using EXT-*AL*. To produce stable and spherical AgNPs, various EXT-*AL* concentrations and solution pH values were adjusted to optimize the synthesis approach. The optimal conditions for green synthesis were identified as an EXT-*AL* concentration of 100 µg/mL and a pH of 10.0, yielding 36.0%. The kinetic growth of AgNPs followed an exponential behavior, with an apparent constant rate of 2.52 × 10^−3^ min^−1^, consistent with previously reported growth rates. Notably, the AgNPs showed excellent antibacterial activity against *S. aureus* and *E. coli*, with bactericidal effects observed at an AgNPs concentration of 3.9 μg/mL (equivalent to Ag^+^). In vitro assays indicated that AgNPs may be biocompatible at lower concentrations, as the viability of HeLa, T731, CaCo-2, and HaCaT remained unaffected at 3.9 μg/mL. However, significant cytotoxic effects were observed at AgNP concentration of ≥ 7.8 μg/mL after 24 h of incubation. It is important to highlight that the negligible cytotoxic effect and strong antibacterial activity observed at low AgNP concentrations underscore their potential for pharmaceutical and biomedical applications.

On the other hand, this study identifies specific challenges, particularly concerning the stability of AgNPs in complex media and potential side effects in living systems. Future research should prioritize the development of surface modification strategies to enhance their colloidal stability of AgNPs and minimize undesired biological interactions. In addition, in vivo studies are essential to better understand their pharmacokinetics, biodistribution, and toxicity profiles. Further evaluations in models of respiratory bacterial infections, particularly those involving clinically relevant strains with well-characterized virulence and antibiotic resistance phenotypes, will be critical to fully assess their therapeutic potential. Additionally, further mechanistic studies, such as ultrastructural analyses, membrane integrity assays, or oxidative stress evaluations, are necessary to elucidate the precise antimicrobial action of the AgNPs and to better understand their interaction with bacterial cells.

## Figures and Tables

**Figure 1 pharmaceutics-17-00672-f001:**
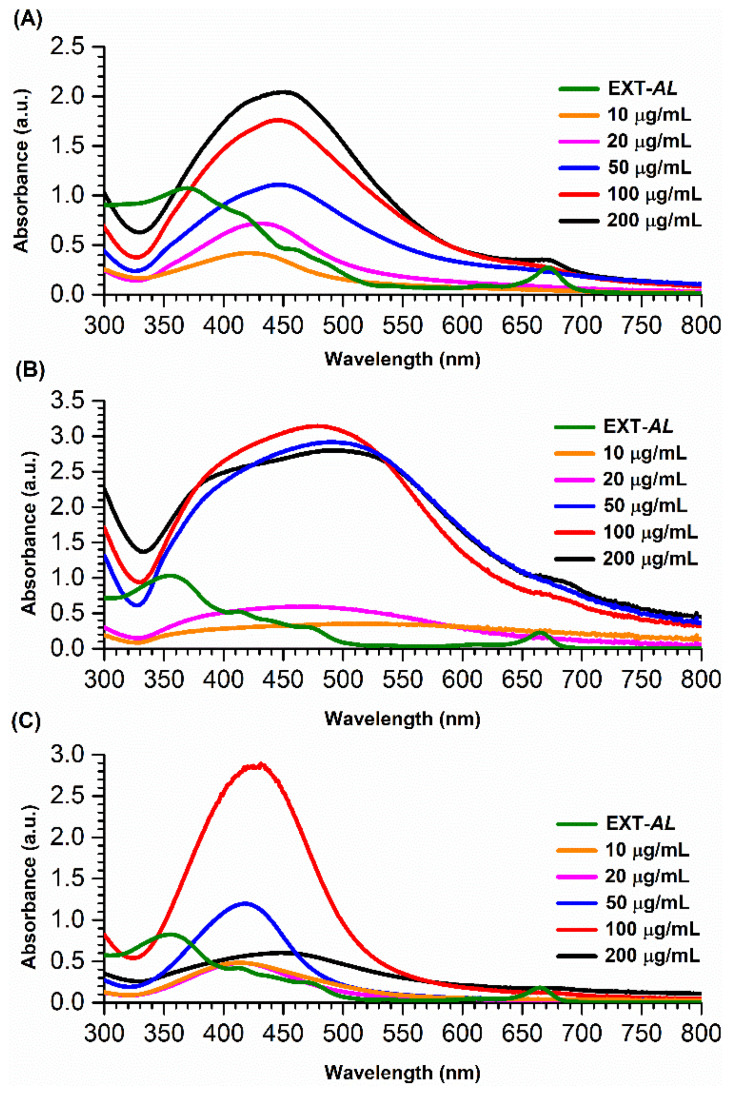
UV-Vis absorption spectra of AgNPs synthesized in various pH conditions: 5.5 (**A**), 8.0 (**B**), and 10.0 (**C**), at different concentrations of *Antigonon leptopus* ethanolic extract.

**Figure 2 pharmaceutics-17-00672-f002:**
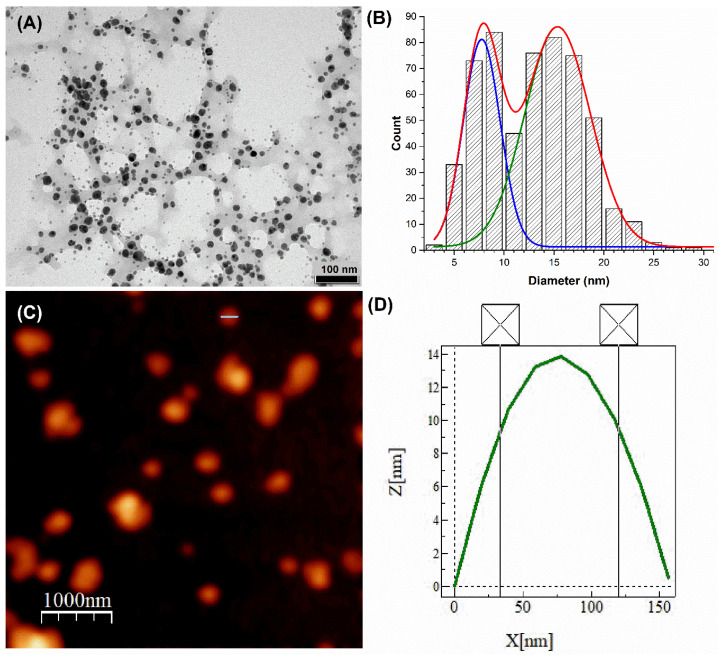
Transmission electron microscopy (TEM) micrograph of AgNPs synthesized at a 100 µg/mL concentration of EXT-*AL* (**A**), size distribution of AgNPs as determined by TEM with Gaussian fitting curves (blue, green, and red represent the average distribution particle size) (**B**), atomic force microscopy (AFM) 2-D topography of AgNPs at 100 µg/mL concentration of EXT-*AL* (**C**), and the height profile along the white line in image C (**D**).

**Figure 3 pharmaceutics-17-00672-f003:**
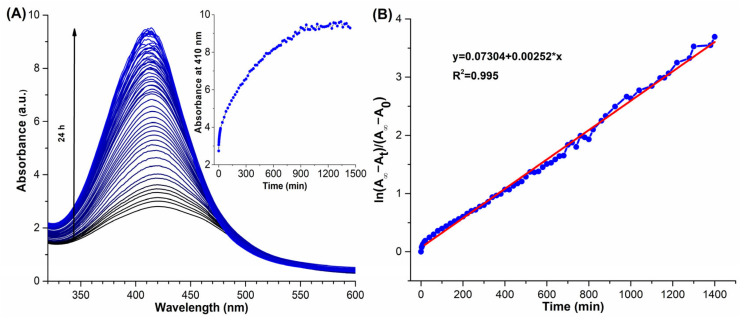
(**A**) UV–Vis spectral evolution of the synthesized AgNPs over 24 h. The color change from black to blue indicates the time progression during nanoparticle formation, with black lines representing early time points and blue lines representing later stages. The inset graph shows the increase in absorbance at 410 nm over time, confirming the formation of AgNP colloids. (**B**) Linear plot of ln((A_∞_ − A_t_)/(A_∞_ − A_0_)) versus time, used to determine the apparent rate constant of the reaction. The red line represents the linear regression fit (R^2^ = 0.995).

**Figure 4 pharmaceutics-17-00672-f004:**
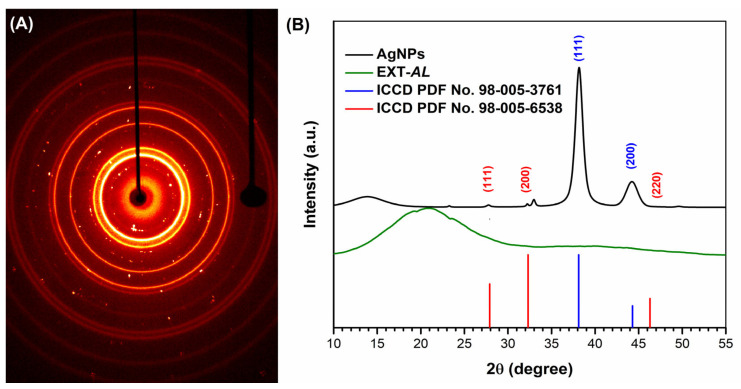
Circular and single-crystal diffraction pattern (**A**) and X-ray diffraction pattern (**B**) of the synthesized AgNPs.

**Figure 5 pharmaceutics-17-00672-f005:**
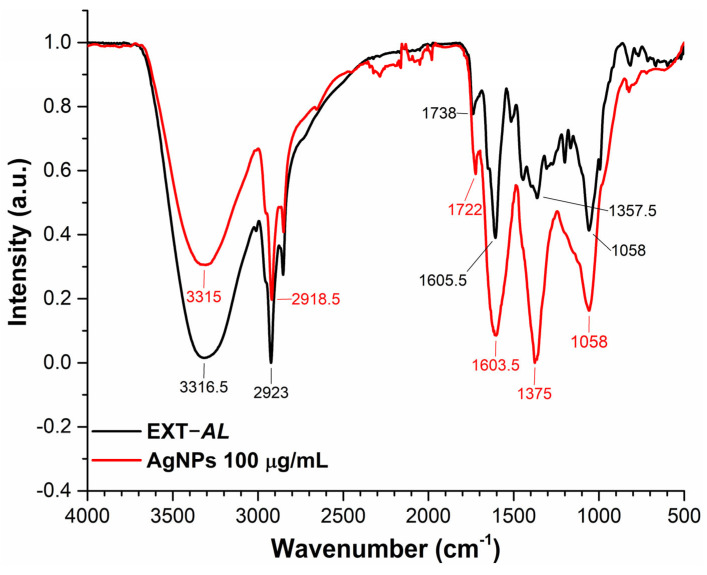
Normalized FTIR spectra of EXT-*AL* (black line) and synthesized AgNPs (red line).

**Figure 6 pharmaceutics-17-00672-f006:**
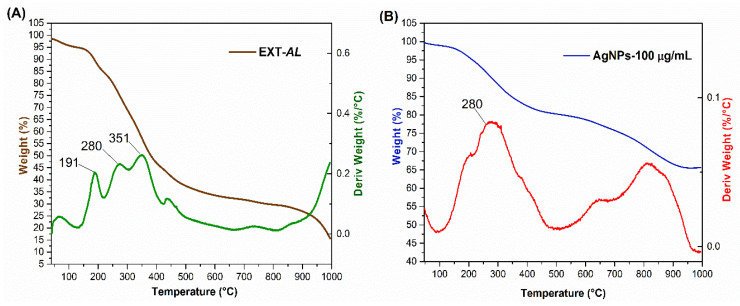
TGA and DTG plots of EXT-*AL* (**A**) and synthesized AgNPs (**B**).

**Figure 7 pharmaceutics-17-00672-f007:**
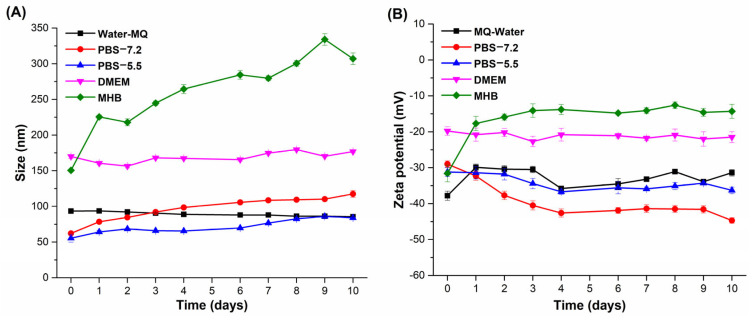
Colloidal stability of AgNPs over 10 days in different dispersion media. (**A**) Hydrodynamic diameter and (**B**) zeta potential measured in: Milli-Q water (black squares), PBS at pH 7.2 (red circles), PBS at pH 5.5 (blue upward triangles), DMEM (magenta downward triangles), and Mueller–Hinton Broth (MHB, green diamonds).

**Figure 8 pharmaceutics-17-00672-f008:**
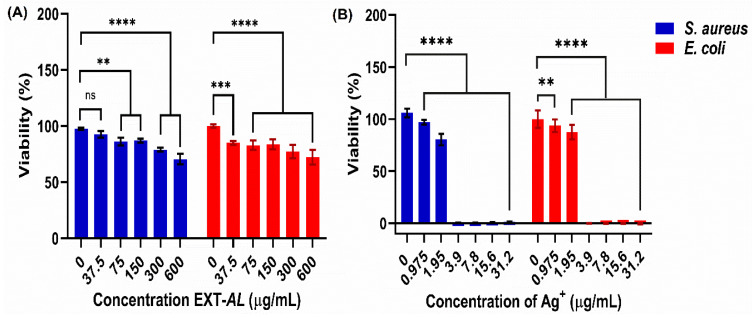
Effect of EXT-*AL* (**A**) and AgNPs (**B**) on the viability of *S. aureus* and *E. coli*. Represented statistical differences between groups are denoted by ns for *p* < 0.1234, ** for *p* < 0.002, *** for *p* < 0.0002, and **** for *p* < 0.0001.

**Figure 9 pharmaceutics-17-00672-f009:**
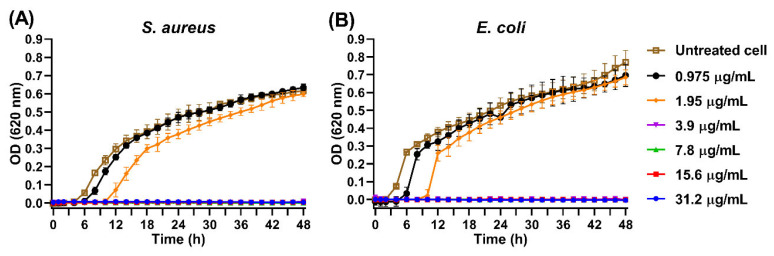
Growth kinetics of *S. aureus* (**A**) and *E. coli* (**B**) in the presence of varying concentrations of AgNPs (0.975–31.2 µg/mL), monitored by optical density at 620 nm over 48 h.

**Figure 10 pharmaceutics-17-00672-f010:**
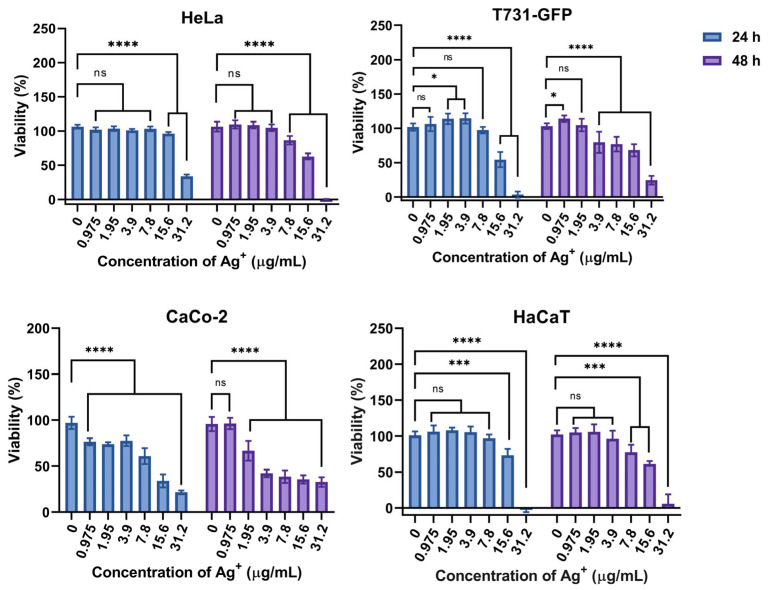
Cellular viability at different AgNPs-100 μg/mL concentrations against HeLa, T731-GFP, CaCo-2, and HaCaT. Represented statistical differences between groups are shown by ns for *p* < 0.1234, * for *p* < 0.033, *** for *p* < 0.0002, and **** for *p* < 0.0001.

**Table 1 pharmaceutics-17-00672-t001:** Hydrodynamic diameter, polydispersity index, and zeta potential of AgNPs obtained at pH 10 and different concentrations of EXT-*AL*.

Concentration of EXT-*AL*	D_h_ (nm)	PDI	ZP (mV)
10 µg/mL	95.10 ± 1.30 ^b^	0.34 ± 0.01 ^c^	−27.88 ± 1.88 ^b^
20 µg/mL	120.10 ± 2.12 ^c^	0.30 ± 0.10 ^c^	−31.90 ± 1.91 ^b^
50 µg/mL	130.90 ± 1.30 ^d^	0.30 ± 0.02 ^b^	−30.50 ± 1.31 ^c^
100 µg/mL	93.48 ± 1.88 ^e^	0.25 ± 0.01 ^a^	−37.80 ± 1.28 ^d^
200 µg/mL	267.90 ± 7.92 ^a^	0.37 ± 0.10 ^ab^	−26.60 ± 0.90 ^a^

D_h_ = hydrodynamic diameter in nanometers, PDI = polydispersity index, ZP = zeta potential. Different superscript letters indicate statistically significant differences between groups, D*_h_*, PDI, and ZP (one-way ANOVA followed by Tukey’s post hoc test, *p* < 0.05).

## Data Availability

All data supporting the findings of this study are included in the present manuscript. Further inquiries can be directed to the corresponding author.

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
