# Peer review of "Phytosynthesis and Characterization of Silver Nanoparticles from Antigonon leptopus: Assessment of Antibacterial and Cytotoxic Properties"

_pharmaceutics, 2025, doi:10.3390/pharmaceutics17050672_

Round 1
Reviewer 1 Report
Comments and Suggestions for Authors
The article “Phytosynthesis and Characterization of Silver Nanoparticles from Antigonon leptopus: Assessment of Antibacterial and Cytotoxic Properties” presents studies concerning the synthesis conditions and the physicochemical and biological properties of silver nanoparticles.
The authors have obtained interesting results that meet the requirements of Pharmaceutics, so, in my opinion, this work can be accepted after careful revision in accordance with the following comments:
- Lines 119-130.
The text provided here is more appropriate for the section “3. Results” or “4. Discussion”, rather than the Introduction, which should demonstrate the scientific novelty of this study.
- Lines 253-255
«FTIR spectra were recorded at the end of the synthesis process to confirm the presence of the EXT-AL compounds adsorbed onto the surface of AgNPs, acting as stabilizing agents.»
This phrase of the authors indicates the formation of the conjugate AgNPs@EXT-AL, despite the fact that the text speaks about the synthesis and biological activity of AgNPs only. In fact, the object of research is a hybrid material in which the core is AgNPs and the shell is EXT-AL.
The TGA data speak in favor of the formation of the conjugate, which indicate that for AgNPs there is a mass loss of more than 30%. If TGA is carried out in an oxygen atmosphere, the mass loss will be even more significant.
- 3.4. X-ray diffraction (XRD)
Lines 443-446
… the presence of organic components from EXT-AL, such as chlorargyrite and magnesium silver,…
Here are shown inorganic compounds that may be present in EXT-AL. In this regard, it is advisable to present in Figure 4, along with the AgNPs diffraction pattern, the EXT-AL diffraction pattern.
The diffractogram in Fig. 4 is very noisy and does not allow us to unambiguously interpret the presence of silver only in the Ag 0 state. It is possible that there is also a certain amount of silver oxide, which can be formed by the interaction of silver nitrate with alkali and be X-ray amorphous.
If possible, it would be good to provide the XPS spectrum of Ag 3d and its analysis for the AgNPs@EXT-AL system, since this can unambiguously show the electronic state of silver. This can show the relationship between the state of the metal and its biological activity.
- The References list is carelessly written, for example:
Some references list all authors (23 and 38), while others list only one author (21 and 30)
Ref. 31, 32 – journal not listed
Ref. 53: «… J. Inorg. Organomet. Polym. Mater., vol. 0, no. 0, p. 0, 2018,…».
Author Response
Comments1: -Lines 119-130.
The text provided here is more appropriate for the section “3. Results” or “4. Discussion”, rather than the Introduction, which should demonstrate the scientific novelty of this study.
Response 1: We appreciate the reviewer’s observation. In response, we have revised the Introduction section to remove detailed descriptions of experimental results, which have now been relocated to the appropriate sections of the manuscript. The revised paragraph now emphasizes the scientific context, the gap in current knowledge, and the novelty of our approach. Specifically, we highlight the lack of studies evaluating how synthesis conditions influence the physicochemical properties and biological performance of phytosynthesized AgNPs. The updated version is as follows (lines 126–132):
Biosynthesis of AgNPs has been widely explored as an eco-friendly alternative to conventional chemical and physical methods. However, limited research has explored how specific synthesis parameters, such as pH, extract concentration, and reaction conditions, affect the physicochemical properties of the resulting nanoparticles. These characteristics, including particle size, shape, and colloidal stability, are critical as they directly influence the biological activity and potential applications of AgNPs, particularly their antimicrobial efficacy and cytotoxicity.
Comments 2: -Lines 253-255
«FTIR spectra were recorded at the end of the synthesis process to confirm the presence of the EXT-AL compounds adsorbed onto the surface of AgNPs, acting as stabilizing agents.»
This phrase of the authors indicates the formation of the conjugate AgNPs@EXT-AL, despite the fact that the text speaks about the synthesis and biological activity of AgNPs only. In fact, the object of research is a hybrid material in which the core is AgNPs and the shell is EXT-AL.
The TGA data speak in favor of the formation of the conjugate, which indicate that for AgNPs there is a mass loss of more than 30%. If TGA is carried out in an oxygen atmosphere, the mass loss will be even more significant.
Response 2: We thank the reviewer for this comment. In response, we have revised the manuscript to clarify that the synthesized nanomaterial corresponds to a hybrid system, AgNPs@EXT-AL, consisting of a metallic silver core stabilized by a shell of biomolecules from the EXT-AL. This is now explicitly stated in the revised FTIR description (Lines 258-261):
FTIR spectra were recorded at the end of the synthesis process to confirm the formation of a complex AgNPs@EXT-AL, where the presence of the EXT-AL compounds adsorbed onto the surface of the AgNPs core, acting as stabilizing agents, forming an organic shell around the AgNPs.
We thank the reviewer for this comment. The TGA results support this conjugate structure. As described in the revised section (lines 524-535):
Figure 6 (B) shows the TGA curve (blue line) of AgNPs, along with the corresponding DTG (red line), highlighting two distinct stages of thermal degradation. The first degradation stage occurs between 150 °C and 450 °C, resulting in a mass loss of approximately 20%. This loss is attributed to the decomposition of the organic coating derived from EXT-AL, which consists of phenolic acids, flavonoids, and carbohydrates [63], [64]. Comparable thermal behavior has been reported for glucose-coated AgNPs, which exhibited a 24% mass loss at 450 °C [36]. Similarly, AgNPs synthesized using green tea phytochemicals demonstrated a 27% mass loss within the same temperature range [36], [65].
A second degradation was observed between 550-900 °C, accounting for an additional 35 % mass loss. This stage is attributed to the calcination of residual organic compounds present in AgNPs. Then, the remaining mass corresponds to silver content, consistent with previous findings reported in the literature [66].
Comments 3: - 3.4. X-ray diffraction (XRD)
Lines 443-446
… the presence of organic components from EXT-AL, such as chlorargyrite and magnesium silver,…
Here are shown inorganic compounds that may be present in EXT-AL. In this regard, it is advisable to present in Figure 4, along with the AgNPs diffraction pattern, the EXT-AL diffraction pattern.
The diffractogram in Fig. 4 is very noisy and does not allow us to unambiguously interpret the presence of silver only in the Ag0 state. It is possible that there is also a certain amount of silver oxide, which can be formed by the interaction of silver nitrate with alkali and be X-ray amorphous.
Response 3: We sincerely thank the reviewer for the valuable observation regarding the interpretation of the X-ray diffraction pattern.
In the revised version of the manuscript, the XRD data were processed to reduce background noise and enhance peak visibility for better interpretation. We have included the diffraction pattern of the EXT-AL extract in Figure 4 B, which allows a clearer distinction of the peaks observed in the AgNPs pattern. These peaks mainly correspond to metallic silver (Ag⁰), with characteristic signals at 2θ ≈ 38.1° and 44.3°, associated with the (111) and (200) planes, respectively, of the face-centered cubic silver structure (ICDD PDF No.98-005-3761).
Although the pattern of EXT-AL presents a certain level of noise, likely due to the organic nature of the plant extract and the low crystallinity of some components, no clear peaks attributable to silver oxides or chlorides were identified.
Also, the methodology section has been updated to indicate that the ICDD PDF database was used to identify crystalline phases in the XRD patterns (line 279). This addition clarifies the approach used to assign diffraction peaks and strengthens the reliability of the structural analysis.
The paragraph was reedited as follows (lines 475-485):
XRD analysis was performed to assess the crystallinity of Ag NPs synthesized using 100 μg/mL EXT-AL at pH 10.0 (Figure 4). The diffraction pattern shown in Figure 4 A confirms the polycrystalline nature of the AgNPs formed under these conditions. Prominent peaks at 2θ 38.1° and 44.1° (Figure 4 B) correspond to the (111) and (200) crystal planes of metallic silver, respectively, features that are characteristic of a FCC structure (ICCD PDF file No: 98-005-3761) [52]. The crystallite size calculated for the (111) plane was approximately 12.3 nm [53], which is consistent with the metallic core size observed in TEM images.
In addition to the main silver peaks, minor diffraction signals were detected that may correspond to inorganic by-products formed during the synthesis process, such as chlorargyrite, which exhibit a cubic crystal structure (ICCD file No: 98-005-6538) [54].
If possible, it would be good to provide the XPS spectrum of Ag 3d and its analysis for the AgNPs@EXT-AL system, since this can unambiguously show the electronic state of silver. This can show the relationship between the state of the metal and its biological activity.
We sincerely thank the reviewer for the valuable observation regarding the suggestion to include the XPS spectrum for the AgNPs.
However, this analysis was not performed in the present study due to instrumental limitations. Nevertheless, the presence of metallic silver (Ag⁰) is supported by the match with the diffraction database (ICDD PDF No.98-005-3761), which is consistent with systems containing silver in its zero-valent state.
Comments 4: - The References list is carelessly written, for example:
while Some references list all authors (23 and 38), others list only one author (21 and 30)
Ref. 31, 32 – journal not listed
Ref. 53: «… J. Inorg. Organomet. Polym. Mater., vol. 0, no. 0, p. 0, 2018,…».
Response 4: We appreciate your valuable comments regarding the references. All observations related to the bibliography have been addressed, and the necessary corrections and updates have been made accordingly.
Note: A Word file named “pharmaceutics-3624929_Highlighted_Changes.docx” has been included with this submission. In this file, all modifications made to the manuscript are highlighted in yellow for easier identification.

Reviewer 2 Report
Comments and Suggestions for Authors
This paper describes the green biosynthesis of AgNPs using Antigonon leptopus leaf extract as both reducing and stabilizing agent. The synthesis was optimized under different pH conditions and extract concentrations, and the resulting AgNPs were tested for antibacterial activity against Escherichia coli and Staphylococcus aureus, as well as cytotoxicity against various cell lines. Overall, this is a well-structured research paper with comprehensive methodology and extensive data. The research content shows novelty and potential application value. However, there are several areas that require improvement, including experimental design, data analysis, and result interpretation.
Major Comments:
- While pH is identified as a key factor in optimizing AgNPs synthesis, the rationale for selecting specifically 5.5, 8.0, and 10.0 is not thoroughly explained. The authors explain their reasoning for choosing these particular pH values and consider testing additional pH values to determine optimal conditions.
- There are noticeable differences in particle sizes observed by TEM and AFM. The authors should provide a more in-depth analysis and discussion of this phenomenon and explain possible causes.
- Figure 7 clearly shows aggregation of AgNPs in PBS, DMEM, and MHB. While common for nanoparticles, the manuscript needs a more thorough discussion on the potential impact of this aggregation on the measured biological activities. Aggregation alters effective concentration, surface area, and nanoparticle-cell interactions.
- The discussion of antimicrobial mechanisms lacks depth. Although the authors mention that AgNPs may act by disrupting bacterial cell walls, direct experimental evidence supporting this claim is missing. I recommend using transmission electron microscopy to observe morphological changes in bacteria after AgNPs treatment, or employing other methods (such as membrane integrity tests or ROS detection) to investigate the antimicrobial mechanism.
- The study shows low toxicity at lower concentrations but significant toxicity at higher concentrations (≥ 7.8 or 15.6 µg/mL, depending on cell line/time). The manuscript needs a clearer discussion of the "therapeutic window" – the relationship between the effective antimicrobial concentration (MIC = 3.9 µg/mL) and the concentration causing cytotoxicity.
Minor Comments:
- The introduction provides a detailed description of Antigonon leptopus characteristics and traditional medicinal value, but lacks sufficient discussion of its potential for AgNPs synthesis. I suggest adding more references and discussion on related literature.
- Clarify if the MIC determination followed a standard serial dilution method as per CLSI guidelines. The description is slightly brief.
- The formula for calculating % AgNPs yield (Eq 4) uses "Initial Concentration of AgNO3 by ICP-MS" as the denominator. This seems incorrect or ambiguously worded. It should likely be the theoretical initial Ag+ concentration based on the added AgNO3 mass or a measured concentration from a dissolved AgNO3 standard. Please verify and clarify/correct.
- The captions for Figures 2 and 9 are relatively simple; more detailed descriptions would be helpful.
- Table 1 lacks statistical analysis results for the data presented.
- In Figure 4,the peaks marked (*) indicating potential impurities/organic components should be discussed more explicitly in the figure legend and text regarding their possible origins. A clearer comparison with a standard Ag JCPDS pattern would be beneficial.
- Figure 7 legend should more clearly label which symbol corresponds to which medium.
- The comparison of current results with other studies in the discussion section is not comprehensive enough. I recommend including comparisons with AgNPs synthesized using other plant extracts to highlight the innovation of this research.
- There are some grammatical and spelling errors in the text. For example, in line 56, "Among the vast arrays of nanoparticles," the expression "vast arrays" is not accurate.
- Some paragraphs are excessively long. I suggest appropriate paragraph breaks to improve readability.
Author Response
Major Comments:
Comments 1: While pH is identified as a key factor in optimizing AgNPs synthesis, the rationale for selecting specifically 5.5, 8.0, and 10.0 is not thoroughly explained. The authors explain their reasoning for choosing these particular pH values and consider testing additional pH values to determine optimal conditions.
Response 1: We thank the reviewer for this insightful observation. The following paragraph was added to the result section (lines 356-363):
The pH values of 5.5, 8.0, and 10.0 were selected based on the known pKa range of phenolic compounds present in EXT-AL, which typically lie between 8.0 and 10.0. At pH 5.5, most phenolic hydroxyl groups remain protonated, limiting their electron-donating capacity and, consequently, their ability to reduce Ag+ ions. In contrast, at pH 8.0 and especially at pH 10.0, increased deprotonation enhances the reducing power of compounds and improves nanoparticle stabilization. Therefore, these pH conditions were systematically evaluated to explore how the ionization state of phenolics influences the efficiency of AgNPs and the stability of the resulting colloidal suspension.
Comments 2: There are noticeable differences in particle sizes observed by TEM and AFM. The authors should provide a more in-depth analysis and discussion of this phenomenon and explain possible causes.
Response 2: We appreciate the reviewer’s insightful comment. The observed differences in particle sizes between TEM and AFM arise from the fundamental differences in the measurement principles of these techniques. AFM analyzes the topographic surface of nanoparticles, capturing the thickness of the organic layer from EXT-AL adsorbed on the AgNPs surface. In contrast, TEM only shows the metallic core, as this organic coating is transparent to the electron beam.
The following paragraph was added at the end of 3.2 Transmission Electron Microscopy (TEM) and Atomic Force Microscopy (AFM) analysis (lines 452-456):
These discrepancies in particle size arise from the fundamental differences between the techniques. AFM measures surface topography, capturing the surrounding organic layer derived from EXT-AL adsorbed onto the AgNPs surface. In contrast, TEM provides high resolution images of the metallic core, as the organic coating is typically transparent to the electron beam.
Comments 3: Figure 7 clearly shows aggregation of AgNPs in PBS, DMEM, and MHB. While common for nanoparticles, the manuscript needs a more thorough discussion on the potential impact of this aggregation on the measured biological activities. Aggregation alters effective concentration, surface area, and nanoparticle-cell interactions.
Response 3: We thank the reviewer for this valuable comment. In response, we have expanded the discussion in the manuscript to address the role of aggregation in influencing the biological activity of AgNPs. In the results and discussion. Firstly, in section 3.7 Stability of AgNPs, the following paragraphs were added (lines 540-555):
The stability of AgNPs synthesized with 100 μg/mL EXT-AL at pH 10.0 was evaluated in different media, including Mili-Q water, PBS buffer (at pH 7.2 and 5.5), DMEM, and MHB. Figures 7 A and 7 B show the size and ZP of AgNPs in the tested media.
In Mili-Q water, the size of AgNPs progressively decreases from 93 to 85 nm over 10 days. In this regard, the surface charge of AgNPs changes from -38 mV to -32 mV during this period, indicating a strong electrostatic repulsion that supports the aqueous stability of the nanoparticles [66].
In PBS buffer at pH 7.2, the size of AgNPs increased from 62 to 118 nm, while the ZP value decreased from -29 mV to -45 mV. In contrast, at pH 5.5, the AgNPs size increased from 56 nm to 84 nm at pH 5.5, with the ZP value decreasing from -31 mV to -36 mV.
In DMEM culture medium supplemented with fetal albumin, the size of AgNPs increased due to the protein adsorption onto their surface, forming large aggregates [67]. The size of AgNPs recorded by DLS was around 170 nm with a ZP of -20 mV, remaining almost constant over the 10-day incubation period. This behavior differs from that observed in the MHB culture medium, where the size increased from 150 nm to 300 nm after 10 days, while the ZP values rose from -32 mV to approximately -14 mV.
And in the discussion section (lines 660-674):
The interaction of AgNPs with biological systems is highly dependent on their colloidal stability and dispersion in physiological environments. To predict their behavior and performance in subsequent biological assays, the stability of AgNPs was evaluated in water, PBS, DMEM, and MHB.
In PBS (pH 7.2 and 5.5), the observed increase in particle size and changes in ZP suggest that phosphate ions induce nanoparticle aggregation by screening surface charges, thereby reducing repulsive forces [79]. In DMEM, the larger hydrodynamic size are primarily attributed to the adsorption of serum proteins, such as fetal bovine serum albumin, onto the AgNPs surface, leading to the formation of large aggregates [67]. Similarly, in MHB, the high ionic strength resulting from Na⁺, K⁺, and Ca²⁺ destabilizes the colloidal system and promotes significant aggregation over time [35].
Although aggregation may enhance sedimentation and reduce the availability of dispersed AgNPs, thereby limiting their effective interactions with biological targets, the synthesized nanoparticles still showed acceptable stability in Milli-Q water, PBS, and DMEM. This is particularly relevant for biomedical applications.
Comments 4: The discussion of antimicrobial mechanisms lacks depth. Although the authors mention that AgNPs may act by disrupting bacterial cell walls, direct experimental evidence supporting this claim is missing. I recommend using transmission electron microscopy to observe morphological changes in bacteria after AgNPs treatment or employing other methods (such as membrane integrity tests or ROS detection) to investigate the antimicrobial mechanism.
Response 4: We appreciate the reviewer’s suggestion regarding the use of TEM to provide direct evidence of bacterial membrane disruption. In fact, we attempted to evaluate the interaction between AgNPs and bacterial cells by TEM during a research stay at the University of Santiago de Compostela. However, we encountered technical limitations related to the fixation process with glutaraldehyde. Specifically, the grids became obstructed by residual material from the bacterial samples, which prevented clear imaging and analysis. Unfortunately, as access to the TEM facility was limited to the duration of the research stay, we currently do not have immediate access to repeat the experiment under optimized conditions.
We agree that a deeper mechanistic investigation would strengthen the manuscript, and we have now acknowledged this limitation and included it as a future direction in the conclusions. Additionally, we will consider complementary methods such as membrane integrity assays or ROS detection in future work to better elucidate the antimicrobial mechanism of our AgNPs.
The paragraph was added at the end of the conclusions (lines 734-740):
Further evaluations in models of respiratory bacterial infections, particularly those involving clinically relevant strains with well-characterized virulence and antibiotic resistance phenotypes, will be critical to fully assess their therapeutic potential. Additionally, further mechanistic studies, such as ultrastructural analyses, membrane integrity assays, or oxidative stress evaluations, are necessary to elucidate the precise antimicrobial action of the AgNPs and to better understand their interaction with bacterial cells.
Comments 5: The study shows low toxicity at lower concentrations but significant toxicity at higher concentrations (≥ 7.8 or 15.6 µg/mL, depending on cell line/time). The manuscript needs a clearer discussion of the "therapeutic window" – the relationship between the effective antimicrobial concentration (MIC = 3.9 µg/mL) and the concentration causing cytotoxicity.
Response 5: We appreciate the reviewer’s valuable observation. In response, we have expanded the discussion to clarify the therapeutic window of the AgNPs. Specifically, we emphasize that the AgNPs displayed strong antibacterial activity at 3.9 µg/mL, a concentration that did not significantly affect the viability of various cell lines, including HeLa, T731, CaCo-2, and HaCaT. Cytotoxic effects were only observed at ≥ 7.8 µg/mL after 24 h exposure, indicating a favorable therapeutic margin for short-term or localized applications.
This information is stated as follows (lines 687-706):
In terms of biocompatibility, AgNPs demonstrate selective cytotoxicity depending on the human cell line and exposure duration. For instance, after 24 and 48 hours of incubation with 31.2 µg/mL AgNPs, cell viability decreased to 31%-5% in HeLa cells and 21%-33% in CaCo-2 cells. These findings are consistent with Jadhav et al. (2018), who observed a 65% reduction in HeLa cell viability after 24 hours of exposure to 39.31 µg/mL AgNPs [83]. In a similar study, Chadive et al. (2024) reported cytotoxic effects on HeLa cells at 40 µg/mL after 48 hours [84].
AgNPs exhibited marked cytotoxic toward T731-GFP and HaCaT cells, with both cell lines showing minimal viability after 24 hours of exposure to 31.2 µg/mL. Notably, T731-GFP cells demonstrated partial recovery after 48 hours, reaching approximately 24% viability. In contrast, HaCaT cells remained sensitive with limited recovery over the same period. Rolin et al. (2019) reported significant cytotoxicity effect only at concentrations exceeding 50 µg/mL [36], suggesting that the AgNPs synthesized in this study exert stronger cytotoxicity at comparatively lower concentrations.
These results highlight the promising potential of EXT-AL-derived AgNPs as antimicrobial biomaterials, particularly for localized biomedical applications where precise dose control is feasible. Nevertheless, the observed time-dependent cytotoxicity underscores the need to carefully optimize both concentration and exposure duration. Comprehensive biocompatibility assessments using advanced in vitro and in vivo models are essential to establish a robust safety profile.
Minor Comments:
Comments 6: The introduction provides a detailed description of Antigonon leptopus characteristics and traditional medicinal value but lacks sufficient discussion of its potential for AgNPs synthesis. I suggest adding more references and discussion on related literature.
Response 6: Thank you for your constructive feedback. I appreciate your suggestion to expand the discussion of Antigonon leptopus' potential for AgNPs synthesis. However, due to the current focus and length constraints of the manuscript, it is challenging to incorporate additional references or further elaboration on this topic while maintaining a balance and concise presentation.
I appreciate your understanding and will certainly consider this point in future revisions, if appropriate.
Comments 7: Clarify if the MIC determination followed a standard serial dilution method as per CLSI guidelines. The description is slightly brief.
Responds 7: Thank you for your observation. We confirm that the MIC determination was performed using a standard broth microdilution method according to the Clinical and Laboratory Standards Institute (CLSI) guidelines (document M07-A10). We have revised section 2.14. Antibacterial activity of AgNPs, to clarify this point and ensure that the protocol description aligns more explicitly with the CLSI standard (lines 303-310).
2.14. Antibacterial activity of AgNPs
The antibacterial activity of EXT-AL and AgNPs was evaluated using the microdilution plate method following Clinical Laboratory Standard Institute (CLSI) protocols (document M07-A10) [50], with Escherichia coli (ATCC 25922) and Staphylococcus aureus (ATCC 25923) as bacterial models. For the EXT-AL, 100 µL of each concentration (37.5, 75, 150, 300, and 600 µg/mL) was added to a 96-well plate (Costar, Corning), followed by the addition of 10 µL of bacterial inoculum (5X105 CFU/mL) prepared in Mueller-Hinton broth (MHB) to each well.
Comments 8:The formula for calculating % AgNPs yield (Eq 4) uses "Initial Concentration of AgNO3 by ICP-MS" as the denominator. This seems incorrect or ambiguously worded. It should likely be the theoretical initial Ag+ concentration based on the added AgNO3 mass or a measured concentration from a dissolved AgNO3 standard. Please verify and clarify/correct.
Response 8: We appreciate the reviewer’s helpful observation. We agree that the original wording was unclear. To address this, we revised the formula in Eq. 4 to more accurately reflect the quantification method used. Specifically, both the silver content in the synthesized AgNPs and the initial Ag⁺ concentration from AgNO₃ were quantified by ICP-MS using digested samples. Therefore, the formula now reads (line 301):
% AgNPs yield=[(Ag+ quantified of sample by ICP-MS)/( Ag+ quantified from AgNO3 by ICP-MS )]∙100 (4)
This correction has been implemented in the manuscript to clarify that both values were experimentally measured rather than theoretically estimated (section 2.13, Inductively coupled plasma-mass spectrometry (ICP-MS), lines 293-301).
Comments 9: The captions for Figures 2 and 9 are relatively simple; more detailed descriptions would be helpful.
Response 9: We appreciate the reviewer’s suggestion regarding the figure titles. As recommended, the titles have been revised to provide more clarity and detail. Specifically:
Figure 2 is now titled (lines 437-440): “Figure 2. Transmission electron microscopy (TEM) micrograph of AgNPs synthesized at a 100 µg/mL concentration of EXT-AL (A), size distribution of AgNPs as determined by TEM (B), atomic force microscopy (AFM) 2-D topography of AgNPs at 100 µg/mL concentration of EXT-AL (C), and the height profile along the white line in image C (D).”
Figure 9 is now titled (lines 592-593): “Figure 9. Growth kinetics of S. aureus (A) and E. coli (B) in the presence of varying concentrations of AgNPs (0.975–31.2 µg/mL), monitored by optical density at 620 nm over 48 h.”
Comments 10: Table 1 lacks statistical analysis results for the data presented.
Response 10: Thank you for your valuable suggestion. We have revised the statistical analysis section to provide a more concise summary of the key findings while retaining all relevant statistical values (Table 1), line 416.
Table 1. Hydrodynamic diameter, polydispersity index, and zeta potential of AgNPs obtained at pH 10 and different concentrations of EXT-AL. |
|||
Concentration of EXT-AL |
Dh (nm) |
PDI |
ZP (mV) |
10 µg/mL |
95.10 ± 1.30b |
0.34 ± 0.01c |
-27.88 ± 1.88b |
20 µg/mL |
120.10 ± 2.12c |
0.30 ± 0.10c |
-31.90 ± 1.91b |
50 µg/mL |
130.90 ± 1.30d |
0.30 ± 0.02b |
-30.50 ± 1.31c |
100 µg/mL |
93.48 ± 1.88e |
0.25 ± 0.01a |
-37.80 ± 1.28d |
200 µg/mL |
267.90 ± 7.92a |
0.37 ± 0.10ab |
-26.60 ± 0.90a |
Dh = hydrodynamic diameter in nanometers, PDI = polydispersity index, ZP = zeta potential. Different superscript letters indicate statistically significant differences between groups, Dh, PDI, and ZP (one-way ANOVA followed by Tukey's post hoc test, p < 0.05). |
The revised paragraph now reads as follows (lines 417-426):
One-way ANOVA revealed statistically significant differences in the hydrodynamic diameter (Dh), polydispersity index (PDI), and zeta potential (ZP) of AgNPs synthesized with different concentrations of EXT-AL (p < 0.0001). Tukey’s post hoc test confirmed statistically significant differences between groups (p < 0.05), except for the comparison between 10 and 20 µg/mL for ZP (p = 0.5829).
The smallest Dh (93.48 ± 1.88 nm), lowest PDI (0.25 ± 0.01), and most negative ZP (–37.42 ± 1.28 mV) were observed at 100 µg/mL, indicating greater uniformity and colloidal stability. In contrast, 200 µg/mL produced larger AgNPs (267.8 ± 7.92 nm) with lower stability (–26.12 ± 0.90 mV). These results suggest that 100 µg/mL is the optimal concentration for producing small, monodisperse, and electrostatically stable AgNPs.
Comments 11: In Figure 4, the peaks marked (*) indicating potential impurities/organic components should be discussed more explicitly in the figure legend and text regarding their possible origins. A clearer comparison with a standard Ag JCPDS pattern would be beneficial.
Response 11: We sincerely thank the reviewer for the valuable observation regarding the interpretation of the X-ray diffraction pattern and the suggestion to include the XPS spectrum for the AgNPs.
In the revised version of the manuscript, the XRD data were processed to reduce background noise and enhance peak visibility for better interpretation. We have included the diffraction pattern of the EXT-AL extract in Figure 4 B, which allows a clearer distinction of the peaks observed in the AgNPs pattern. These peaks mainly correspond to metallic silver (Ag⁰), with characteristic signals at 2θ ≈ 38.1° and 44.3°, associated with the (111) and (200) planes, respectively, of the face-centered cubic silver structure (ICDD PDF No.98-005-3761).
Although the pattern of EXT-AL presents a certain level of noise, likely due to the organic nature of the plant extract and the low crystallinity of some components, no clear peaks attributable to silver oxides or chlorides were identified.
Also, the methodology section has been updated to indicate that the ICDD PDF database was used to identify crystalline phases in the XRD patterns. This addition clarifies the approach used to assign diffraction peaks and strengthens the reliability of the structural analysis.
The paragraph was reedited as follows (lines 475-485):
XRD analysis was performed to assess the crystallinity of Ag NPs synthesized using 100 μg/mL EXT-AL at pH 10.0 (Figure 4). The diffraction pattern shown in Figure 4 A confirms the polycrystalline nature of the AgNPs formed under these conditions. Prominent peaks at 2θ 38.1° and 44.1° (Figure 4 B) correspond to the (111) and (200) crystal planes of metallic silver, respectively, features that are characteristic of a FCC structure (ICCD PDF file No: 98-005-3761) [52]. The crystallite size calculated for the (111) plane was approximately 12.3 nm [53], which is consistent with the metallic core size observed in TEM images.
In addition to the main silver peaks, minor diffraction signals were detected that may correspond to inorganic by-products formed during the synthesis process, such as chlorargyrite, which exhibit a cubic crystal structure (ICCD file No: 98-005-6538) [54]
Comments 12: Figure 7 legend should more clearly label which symbol corresponds to which medium.
Response 12: Thank you for your observation. The legend of Figure 7 has been revised to explicitly indicate the symbol, color, and corresponding medium to improve clarity for the reader (lines 558-561).
Figure 7. Colloidal stability of AgNPs over 10 days in different dispersion media. (A) Hydrodynamic diameter and (B) zeta potential measured in: Milli-Q water (black squares), PBS at pH 7.2 (red circles), PBS at pH 5.5 (blue upward triangles), DMEM (magenta downward triangles), and Mueller–Hinton Broth (MHB, green diamonds).
Comments 13: The comparison of current results with other studies in the discussion section is not comprehensive enough. I recommend including comparisons with AgNPs synthesized using other plant extracts to highlight the innovation of this research.
Response 13: Thank you for this valuable recommendation. We agree that a broader comparison with AgNPs synthesized from other plant extracts would enrich the discussion. However, in the present work, we focused our bibliographic search on studies using comparable conditions, including the same bacterial strains for MIC determination and the same cell lines for cytotoxicity evaluation. This criterion significantly limited the number of available studies for direct comparison.
Due to these constraints, we were unable to incorporate a more comprehensive comparison in this version. Nonetheless, we acknowledge the importance of this point and plan to address it in future work by broadening the comparative analysis in follow-up studies.
Comments 14: There are some grammatical and spelling errors in the text. For example, in line 56, "Among the vast arrays of nanoparticles," the expression "vast arrays" is not accurate.
Response 14: Thank you for pointing out the inaccuracy in the expression "vast arrays of nanoparticles." We have revised the sentence to improve clarity and scientific accuracy. The phrase has been changed to:
"Among the numerous types of nanoparticles," (line 58)
Comments 15: Some paragraphs are excessively long. I suggest appropriate paragraph breaks to improve readability.
Response 15: Thank you for your valuable feedback. I carefully considered your suggestion regarding the long paragraphs. I addressed this issue as much as the structure and organization of the information allowed. In some cases, the content required maintaining longer paragraphs for clarity and coherence, but I have made paragraph breaks where possible to improve readability. I appreciate your understanding and look forward to any further comments you may have.
Note: A Word file named “pharmaceutics-3624929_Highlighted_Changes.docx” has been included with this submission. In this file, all modifications made to the manuscript are highlighted in yellow for easier identification.

Round 2
Reviewer 1 Report
Comments and Suggestions for AuthorsРецензент удовлетворен ответами авторов на свои вопросы. Статья может быть опубликована в журнале Pharmaceutics

Reviewer 2 Report
Comments and Suggestions for Authors
No